

# A unified simulation model for understanding the diversity of cancer evolution

Atsushi Niida[1], Takanori Hasegawa[2], Hideki Innan[3], Tatsuhiro Shibata[1,6], Koshi Mimori[4] and Satoru Miyano[5]

[1] Laboratory of Molecular Medicine, Human Genome Center, The Institute of Medical Science, The University of Tokyo, Tokyo, Japan
[2] Division of Health Medical Data Science, Health Intelligence Center, The Institute of Medical Science, The University of Tokyo, Tokyo, Japan
[3] SOKENDAI, The Graduate University for Advanced Studies, Hayama, Japan
[4] Department of Surgery, Kyushu University Beppu Hospita, Beppu, Japan
[5] Laboratory of DNA Information Analysis, Human Genome Center, The Institute of Medical Science, The University of Tokyo, Tokyo, Japan
[6] Division of Cancer Genomics, National Cancer Center Research Institute, Tokyo, Japan

Corresponding author
Atsushi Niida,
aniida@ims.u-tokyo.ac.jp

## ABSTRACT

Because cancer evolution underlies the therapeutic difficulties of cancer, it is clinically important to understand the evolutionary dynamics of cancer. Thus far, a number of evolutionary processes have been proposed to be working in cancer evolution. However, there exists no simulation model that can describe the different evolutionary processes in a unified manner. In this study, we constructed a unified simulation model for describing the different evolutionary processes and performed sensitivity analysis on the model to determine the conditions in which cancer growth is driven by each of the different evolutionary processes. Our sensitivity analysis has successfully provided a series of novel insights into the evolutionary dynamics of cancer. For example, we found that, while a high neutral mutation rate shapes neutral intratumor heterogeneity (ITH) characterized by a fractal-like pattern, a stem cell hierarchy can also contribute to shaping neutral ITH by apparently increasing the mutation rate. Although It has been reported that the evolutionary principle shaping ITH shifts from selection to accumulation of neutral mutations during colorectal tumorigenesis, our simulation revealed the possibility that this evolutionary shift is triggered by drastic evolutionary events that occur in a short time and confer a marked fitness increase on one or a few cells. This result helps us understand that each process works not separately but simultaneously and continuously as a series of phases of cancer evolution. Collectively, this study serves as a basis to understand in greater depth the diversity of cancer evolution.

## INTRODUCTION

Cancer is regarded as a disease of evolution; during tumorigenesis, a normal cell evolves to a malignant population by means of mutation accumulation and adaptive Darwinian

selection. Evolution allows cancer cells to adapt to a new environment and acquire malignant phenotypes such as metastasis and therapeutic resistance. Therefore, it is clinically important to understand cancer evolutionary dynamics. The view of cancer as an evolutionary system was established by *Nowell (1976)*. By combining this view with a series of discoveries of onco- and tumor suppressor genes (hereinafter, collectively referred to as "driver genes"), *Fearon & Vogelstein (1990)* proposed a multistep model for colorectal carcinogenesis. Since then, cancer evolution has generally been described as "linear evolution," where driver mutations are acquired linearly in a step-wise manner, generating a malignant clonal population.

However, this simple view of cancer evolution has been challenged since the advent of the next generation sequencing technology (*Yates & Campbell, 2012*; *McGranahan & Swanton, 2017*; *Niida et al., 2018*). Deep sequencing demonstrated that subclonality prevails in both blood and solid tumors, and multiregion sequencing of various types of solid tumor more dramatically unveiled intratumor heterogeneity (ITH), which results from the branching process in a cancer cell population along with mutation accumulation. These genomic studies also found that subclones often harbor mutations in known driver genes, suggesting that at least a part of ITH is subject to Darwinian selection. In some types of cancer, such as renal cell carcinoma (*Turajlic et al., 2018*) and low-grade glioma (*Suzuki et al., 2015*), this Darwinian selection-driven branching process is especially prominent; we observed convergent evolution in which different subclonal mutations are acquired in the same driver gene or pathway.

Other types of tumors, however, show no clear enrichment of driver mutations in subclonal mutations. Consistently with this observation, several studies employing mathematical modeling have suggested that the accumulation of neutral mutations that do not affect the growth or survival of cancer cells mainly shapes ITH; that is, "neutral evolution" is the major contributor of ITH in multiple types of cancers (*Uchi et al., 2016*; *Sottoriva et al., 2015*; *Ling et al., 2015*; *Niida, Iwasaki & Innan, 2019*). The evolutionary principles shaping ITH differ not only among cancer types but also between stages of tumorigenesis. We and others have recently reported that ITH in the early stage of colorectal tumorigenesis involves selection, whereas the accumulation of neutral mutations plays the central role in shaping IHT in the later stages (*Saito et al., 2018*; *Cross et al., 2018*).

In addition to single nucleotide mutation- and small indel-driven drivers, recent studies have demonstrated that, in multiple types of cancers, more drastic chromosome- and/or genome-wide evolutionary events producing copy number alterations and chromosomal rearrangements may have occurred in a short time at the early stage of cancer evolution (*Gao et al., 2016*; *Baca et al., 2013*). Such large-scale events could confer a marked fitness increase on one or a few cells, which expand to constitute the tumor mass uniformly. This type of evolution is referred to as "punctuated evolution" after the term "punctuated equilibrium", which was proposed for species evolution by Gould and Eldredge to challenge the long-standing paradigm of gradual Darwinian evolution (*Gould & Eldredge, 1972*; *Jay Gould & Eldredge, 1993*), although the underlying molecular mechanisms that cause rapid bursts of change are very different.

Collectively, at least four scenarios of cancer evolution were proposed (*Davis, Gao & Navin, 2017*). In this paper, we term the four scenarios as the linear-replacing, punctuated-replacing, driver-branching, and neutral-branching processes (Figs. 1A–1D). The linear-replacing process applies when newly arisen clones repeatedly spread and replace the entire population very quickly. A special case of the linear-replacing process is the punctuated-replacing process, where a number of drastic changes occur in a very short time and a very fit clone spreads and replaces the entire population very quickly. In the driver-branching process, multiple subclones having distinct driver mutations coexist to shape ITH, whereas, in the neutral-branching process, there are no significant driver mutations when accumulating mutations that constitute ITH.

To obtain an understanding of cancer evolutionary dynamics, many mathematical models of cancer evolution have been developed (*Beerenwinkel et al., 2014*; *Altrock, Liu & Michor, 2015*); in particular, agent-based simulation models are commonly employed for this purpose (*Sottoriva et al., 2015*; *Waclaw et al., 2015*; *Uchi et al., 2016*; *Iwasaki & Innan, 2017*; *Minussi et al., 2019*; *Poleszczuk, Hahnfeldt & Enderling, 2015*). In agent-based simulation models, each cell in a tumor corresponds to an agent; the cells can divide to produce new cells, die, or migrate, and each cell's behavior can be stochastically determined from its own state and/or the environment surrounding the cell. By applying sensitivity analysis to the simulation models, (i.e., examining the simulation results while changing the parameters of the models), it is possible to identify the factors affecting the cancer evolutionary dynamics (*Niida, Hasegawa & Miyano, 2019*). However, to the best of our knowledge, there exists no simulation work aiming to reproduce and analyze the four above-stated evolutionary processes in a unified manner.

In this paper, we introduce a unified agent-based simulation model, which is simple but sufficient to reproduce the four evolutionary processes (Figs. 1A–1D). Although the unified model is formulated in 'Materials & Methods', the 'Results' section presents a family of simulation models, each of which constitutes submodels of the unified model. While constructing the submodels, we explore the conditions leading to, and the ITH pattern from the four processes. The 'Results' section is composed of four parts. In the first part, we introduce the driver model, which contains only driver mutations, and examine the conditions leading to the linear-replacing and driver-branching processes. In the second part, the neutral model, which contains only neutral mutations, is introduced to address the conditions leading to a neutral pattern of ITH. We show that, although a high neutral mutation rate is necessary for the neutral pattern of ITH, a stem cell hierarchy can also contribute to the neutral pattern by apparently increasing the mutation rate. In the third part, we present a combination of these two models as a composite model and reproduce realistic ITH patterns, which are generated by mixing the neutral pattern with the pattern from the linear-replacing or driver-branching processes. In the final part, we build the punctuated model by incorporating the punctuated-replacing process into the composite model. Our simulation based on the punctuated model demonstrates that the punctuated-replacing process triggers an evolutionary shift from the driver- to the neutral-branching process that is commonly observed during colorectal tumorigenesis

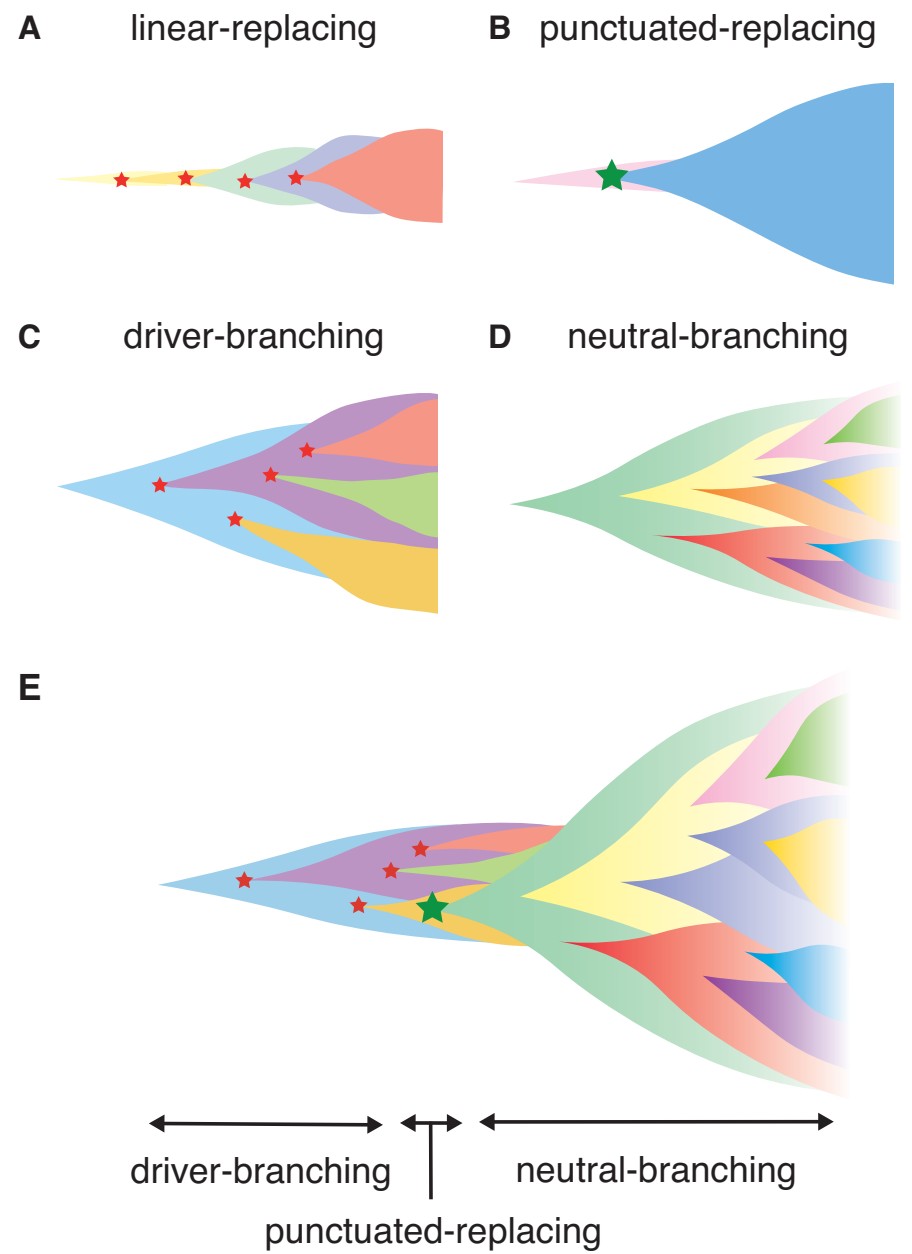

**Figure 1** **Illustrating the scenarios in cancer evolution.** (A–D) The four typical evolutionary processes. Red stars indicate normal driver events, which are assumed to be single nucleotide mutations and small indels, while a green star indicates more drastic chromosome- and/or genome-wide evolutionary events producing copy number alterations and chromosomal rearrangements. (E) Our model explaining the temporal shift of evolutionary principles shaping ITH during colorectal tumorigenesis.

(Fig. 1E). This result helps us understand that each process works not separately but simultaneously and continuously as a series of "phases" of cancer evolution.

## MATERIALS & METHODS

### Simulation model

Although we described a family of simulation models in the Results section, we here formulate the unified model, which encompasses these models. Starting from a stem cell without mutations, the following time steps are repeated until the number of population size $p$ reaches $P$ or the number of time steps $t$ reaches $T$. For each time step, each cell is subject to cell division with a probability $g$ and cell death with a probability $d$. $g$ depends on a base division rate $g_0$, the increase in the cell division probability per driver mutation $f$, the number of driver mutations accumulated in the cell $n_d$, population size $p$, and the carrying capacity $p_c$: $g = g_0 f^{n_d}(1 - p/p_c)$. $d$ depends on the base death rate $d_0$, the decrease in the cell death probability per driver mutation, and the number of driver mutations accumulated in the cell $n_d$: $d = d_0 e^{-n_d}$. When the cell is a differentiated cell, $d_0$ is replaced by $d_0^d$, which is the base death rate for differentiated cells: $d = d_0^d e^{-n_d}$. The order of the trials of cell division and death is flipped with probability 0.5. We also assumed that cell death occurs only in the case where $p > 1$, to prevent the simulation from halting before clonal expansion.

In a cell division, the cell is replicated into two daughter cells. If the parent cell is a stem cell, one of the two daughter cells is differentiated with a probability $1 - s$; that is, $s$ expresses the probability of symmetrical division. For each of the two daughter cells, we introduce $k_d$ driver and $k_n$ neutral mutations. $k_d$ and $k_n$ are sampled from Poison distributions, the parameters of which are $m_d/2$ and $m_n/2$, respectively: $k_d \sim \mathrm{Pois}(m_d/2)$ and $k_n \sim \mathrm{Pois}(m_n/2)$. Note that this means that each cell division generates $m_d$ driver and $m_n$ neutral mutations on average. We assumed each mutation acquired by different division events occurs at different genomic positions and each cell can accumulate $N_d$ driver and $N_n$ neutral mutations at maximum. When each of the two daughter cells has $N_d$ driver mutations, we further attempted to introduce an explosive driver mutation; the explosive driver mutation is introduced with a probability $m_e$ and sets the carrying capacity $p_c$ of the cell to infinite. The pseudocode for the unified model is provided as Algorithm 1. The variables and parameters employed in the unified model are listed in Tables 1 and 2. The simulation code used in this study is available from https://github.com/atusiniida/canevosim.

### Post-processing of simulation results

To evaluate the simulation results quantitatively, we calculated summary statistics based on 1,000 cells randomly sampled from each simulated tumor. these summary statistics are listed in Table 3. time and population size indicate the numbers of time steps and cells, respectively, when the simulation is complete. mutation count per cell represents the mean number of mutations accumulated in each of the randomly sampled 1,000 cells. By combining the mutations of the 1,000 cells, we defined the mutations that occur in 95% or more of the 1,000 cells as clonal mutations, and the others as subclonal mutations.

---

**Algorithm 1** Unified model

---

1: prepare a stem cell without mutations
2: **while** $p < P$ or $t < T$ **do**
3:     **for** each cell **do**
4:         $g = g_0 f^{n_d}(1 - p/p_c)$
5:         $d = d_0 e^{-n_d}$
6:         **if** the cell is a differentiated cell **then**
7:             $d = d_0^d e^{-n_d}$
8:         **if** rand $< 0.5$ **then**
9:             **if** rand$() < g$ **then**
10:                 divide(the cell)
11:             **if** p $> 1$ and rand$() < d$ **then**
12:                 kill the cell (accordingly, $p = p - 1$ )
13:                 # in the case that the cell is replicated, kill one of the two daughter cells
14:         **else**
15:             **if** p $> 1$ and rand$() < d$ **then**
16:                 kill the cell (accordingly, $p = p - 1$)
17:             **if** rand$() < g$ **then**
18:                 divide(the cell)
19:     $t = t + 1$
20:
21:
22: **function** rand$()$
23:     **return** a random number ranging from 0 to 1
24:
25: **function** divide(a cell)
26:     replicate the cell into two daughter cells (accordingly, $p = p + 1$)
27:     **if** the parent cell is a stem cell **then**
28:         **if** rand$() > s$ **then**
29:             differentiate one of the daughter cells
30:     **for** each of the daughter cells **do**
31:         introduce $k_d \sim \mathrm{Pois}(m_d/2)$ driver mutations
32:         introduce $k_n \sim \mathrm{Pois}(m_n/2)$ neutral mutations
33:         **if** $n_d = \sum k_d$ reaches the upper limit $N_d$ **then**
34:             **if** rand$() < m_e$ **then**
35:                 set $p_c$ of the cell to infinite

---

**Table 1  Variables.**

| Symbol | Description |
| --- | --- |
| $k_d$ | Number of driver mutations obtained in a cell division |
| $n_d$ | Number of driver mutations accumulated in a cell |
| $k_n$ | Number of neutral mutations obtained in a cell division |
| $p$ | Population size |
| $t$ | Number of time steps |
| $g$ | Cell division probability |
| $d$ | Cell death probability |

**Table 2  Parameters.**

| Symbol | Description |
| --- | --- |
| $m_d$ | Expected number of driver mutations generated per cell division |
| $m_n$ | Expected number of neutral mutations generated per cell division |
| $m_e$ | Probability of acquiring an explosive mutation |
| $N_d$ | Maximum number of driver mutations accumulated in a cell |
| $N_n$ | Maximum number of neutral mutations accumulated in a cell |
| $f$ | Increase of the cell division probability per driver mutation |
| $e$ | Decrease of the cell death probability per driver mutation |
| $g_0$ | Base cell division probability |
| $d_0$ | Base cell death probability for stem cells |
| $d_0^d$ | Base cell death probability for differentiated cells |
| $s$ | Symmetrical division probability |
| $p_c$ | Carrying capacity |
| $P$ | Maximum population size |
| $T$ | Maximum number of time steps |

The numbers of clonal, subclonal, and both types of mutations were then defined as clonal mutation count, subclonal mutation count, and total mutation count, from which clonal mutation proportion and subclonal mutation proportion were further calculated. The degree of ITH was also measured by Shannon and Simpson indices, which were calculated based on the proportions of different subclones (i.e., cell subpopulations with different mutations) after removing mutations having a frequency less of than 5% or 10%: Shannon index 0.05, Shannon index 0.1, Simpson index 0.05, and Simpson index 0.1. Similarly, after removing mutations having a frequency of less than 5% or 10%, we also checked whether multiple subclones harboring different driver mutations coexist, which is represented as binary statistics, driver-branching 0.05, and driver-branching 0.1. When the simulated tumor had differentiated cells or subclones with explosive driver mutations, the proportion of the subpopulation was calculated as subpopulation proportion.

The single-cell mutation profiles of the 1,000 cells are represented as a binary matrix, the row and column indices of which are mutations and samples, respectively. To interpret the simulation results intuitively, we also visualized the binary matrix by utilizing the

**Table 3  Summary statistics.**

| Name | Description |
|---|---|
| time | Number of time steps when simulation is finished |
| population size | Number of cells when simulation is finished |
| mutation count per cell | Mean number of mutations accumulated in each cell |
| clonal mutation count | Number of clonal mutations |
| subclonal mutation count | Number of subclonal mutations |
| total mutation count | clonal mutation count + subclonal mutation count |
| clonal mutation proportion | clonal mutation count / total mutation count |
| subclonal mutation proportion | subclonal mutation count / total mutation count |
| Shannon index 0.1 | Shannon index calculated with a mutation frequency cutoff of 0.1 |
| Shannon index 0.05 | Shannon index calculated with a mutation frequency cutoff of 0.05 |
| Simpson index 0.1 | Simpson index calculated with a mutation frequency cutoff of 0.1 |
| Simpson index 0.05 | Simpson index calculated with a mutation frequency cutoff of 0.05 |
| driver-branching 0.05 | Binary statistic indicating that multiple subclones harboring different driver mutations coexist, calculated with a mutation frequency cutoff of 0.05 |
| driver-branching 0.1 | Binary statistic indicating that multiple subclones harboring different driver mutations coexist, calculated with a mutation frequency cutoff of 0.1 |
| subpopulation proportion | proportion of differentiated cells or subclones with explosive driver mutations |

heatmap function in R after the following pre-processing, if necessary. When the number of rows was less than 10, empty rows were added to the matrix so that the number of rows was 10. When the number of rows was more than 300, we extracted the 300 rows with the highest mutation occurrence so that the number of rows was 300. In the neutral and neutral-s models, we exceptionally set the maximum row number to 1,000 in order to visualize low-frequency mutations. The visualized matrix is accompanied by a left-side blue bar indicating the driver mutations. When the simulated tumor had differentiated cells or subclones with explosive driver mutations, the subpopulation is indicated by the purple bar on the top of the visualized matrix.

## Sensitivity analysis based on MASSIVE

To cover a sufficiently large parameter space in the sensitivity analysis, we employed a supercomputer, SHIROKANE4 (at Human Genome Center, The Institute of Medical Science, The University of Tokyo). The simulation and post-processing steps for different parameter settings were parallelized on Univa Grid Engine. For each model, we employed a full factorial design involving four parameters (i.e, we tested every combination of candidate values of the four parameters) while other parameters were fixed. The parameter values
used for our analysis are listed in Table 2. For each parameter setting, 50 Monte Carlo trials were performed and the summary statistics were averaged over the 50 trials. The averaged summary statistics calculated for each parameter setting were visualized by interactive heat maps on a web-based visualization tool, the MASSIVE viewer. The MASSIVE viewer also displays single-cell mutation profiles from 5 of the 50 trials with the same parameter setting. For details, please refer to our methodological report (*Niida, Hasegawa & Miyano, 2019*). All the results in this study can be interactively explored in the MASSIVE viewer on our website (https://www.hgc.jp/~niiyan/canevosim). Parameter values used for the MASSIVE analysis are provided in Table S1.

## RESULTS

### Driver model

First, we constructed the "driver" model, which contains only driver genes, aiming to study the two Darwinian selection processes: linear-replacing and driver-branching. We employed an agent-based model where each cell in a tumor is represented by an agent. The model starts from one cell without mutations. In a unit time, a cell divides into two daughter cells with a probability $g$. This model assumes that immortalized cell, which just divides without dying. In each cell division, each of the two daughter cells acquires $k_d$ driver mutations. Here, $k_d$ is sampled from a Poisson distribution with the parameter $m_d/2$, i.e., $k_d \sim \mathrm{Pois}(m_d/2)$, which means that one cell division generates $m_d$ mutations on average. We assumed that driver mutations acquired by different division events occur at different genomic positions and each cell can accumulate $N_d$ mutations at maximum. In this study, we assumed that all mutations are driver mutations, which increase the cell division rate. When the cell acquires mutations, the cell division rate increases $f$ fold per mutation; that is, when a cell has $n_d$ $(= \sum k_d)$ mutations in total, the cell division probability $g$ is defined as $g = g_0 f^{n_d}$, where $g_0$ is a base division probability. In each time step, every cell is subject to a cell division trial, which is repeated until population size $p$ reaches $P$ or the number of time steps $t$ reaches $T$.

To examine the manner in which each parameter affects the evolutionary dynamics of the simulation model, we performed a sensitivity analysis utilizing MASSIVE (*Niida, Hasegawa & Miyano, 2019*), for which we employed a supercomputer. MASSIVE first performs a very large number of agent-based simulations with a broad range of parameter settings. The results are then intuitively evaluated by the MASSIVE viewer, which interactively displays heat maps of summary statistics and single-cell mutation profiles from the simulations with each parameter setting. In Figs. 2A–2C and Fig. S1, the heat maps of three representative summary statistics, the proportion of clonal mutations ( clonal mutation proportion), a measure for ITH ( Shannon index 0.05), and an indicator for the occurrence of the driver-branching process ( driver-branching 0.05), are presented for a part of the parameter space examined. To calculate clonal mutation proportion, we defined the mutations having a frequency of 95% or more as clonal mutations. Shannon index 0.05 is the Shannon index calculated based on the proportions of different subclones (i.e., cell subpopulations with different mutations) after removing the mutations having

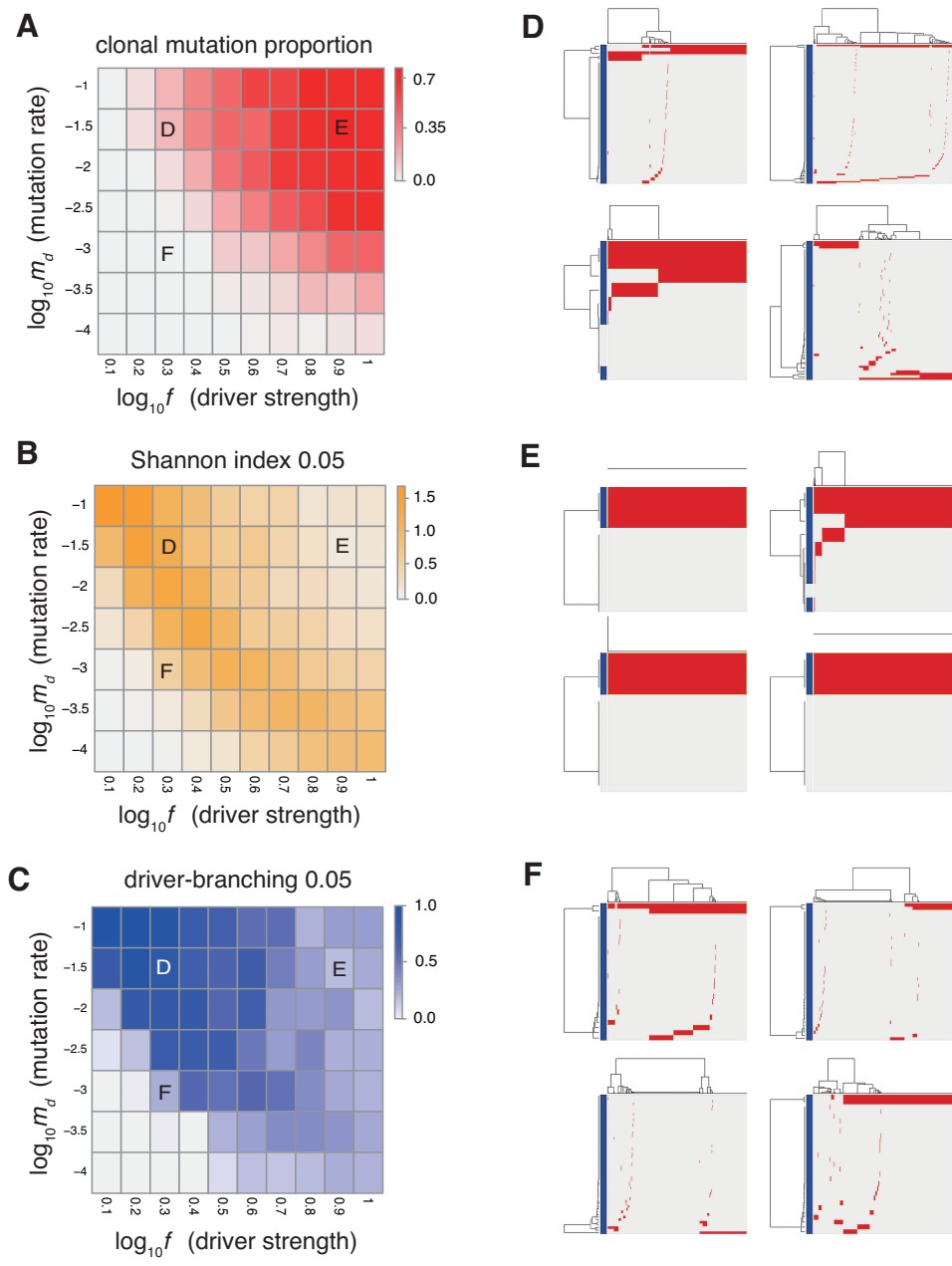

**Figure 2** **Sensitivity analysis of the driver model.** While changing the driver mutation rate $m_d$ and the strength of driver mutations $f$, heat maps of the summary statistics were prepared for the proportion of clonal mutations, clonal mutation proportion (A), a measure for ITH, Shannon index 0.05 (B), and an indicator for the occurrence of the driver-branching process, driver-branching 0.05 (C). $N_d$ and $P$ were set to 3 and $10^5$, respectively. (D–F) Single-cell mutations profiles obtained from four Monte Carlo trials with each of the three parameter settings, which are indicated on the heat maps presented in A–C. Rows and columns of the clustered single-cell mutations profile matrices denote mutations and cells, respectively. Blue side bars indicate driver mutations.

a frequency less of than 5%. The Shannon index is commonly used to measure species richness in community ecology, and it has a positive correlation with diversity. Similarly, after removing mutations having a frequency of less than 5%, we also checked whether multiple subclones harboring different driver mutations coexist, which is represented as a binary statistic, driver-branching 0.05. For each parameter setting, 50 Monte Carlo trials were performed and the summary statistics were averaged over the 50 trials. To examine ITH visually, we sampled 1,000 cells from a simulated tumor and obtained a single-cell mutation profile matrix. The mutation profile matrix was visualized after reordering its rows and columns based on hierarchical clustering. The rows and columns index mutations and samples, respectively (Figs. 2D–2F). All the results can be interactively explored in the MASSIVE viewer on our website (https://www.hgc.jp/~niiyan/canevosim/driver).

The results of the MASSIVE sensitivity analysis demonstrated that the strength of driver mutations $f$ is the most prominent determinant of the Darwinian selection processes (Fig. 2). A smaller value of $f$ (e.g., $f = 10^{0.3}$), which indicates weaker driver mutations, is generally associated with the driver-branching process, which is characterized by large driver-branching 0.05, corresponding to parameter setting D in Figs. 2A–2C. However, in the case of a low mutation rate (e.g., $m_d = 10^{-3}$), a small $f$ value is insufficient to cause expansions of multiple clones, corresponding to parameter setting F in Figs. 2A–2C. When the value of $f$ is large (e.g., $f = 10^{0.9}$), driver-branching 0.05 is small, but the clonal mutation proportion is large, which suggests that the linear-replacing process generates a homogeneous tumor, corresponding to parameter setting E in Figs. 2A–2C. By considering these results with time-course snapshots of the simulations, mechanisms driving the linear-replacing and driver-branching processes were intuitively interpreted (Fig. 3). Under the assumption of weak driver mutations, before a clone that has acquired the first driver mutation becomes dominant, other clones that have acquired different mutations expand, leading to the driver-branching process (Figs. 3A and 3B). In contrast, under the assumption of strong driver mutations, a clone that has acquired the first driver mutation rapidly expands to obtain more driver mutations serially, leading to the linear-replacing process (Figs. 3C and 3D).

The linear-replacing process is very similar to the fixation and selective sweep described in the standard population genetics framework (*Maynard Smith & Haigh, 1974*; *Ohta & Kimura, 1975*). Note that, in a strict sense, fixation does not occur under the assumption that cancer cells are immortal (*Sidow & Spies, 2015*; *Ohtsuki & Innan, 2017*; *Niida, Iwasaki & Innan, 2019*); even if a tumor appears to be monoclonal in a mutation profile for 1,000 randomly sampled cells, it is possible that minor clones having less fitness coexist in the actual population. In the driver-branching process, we observe various subclones that coexist in the population. They could compete with each other depending on their fitness. If different subclones obtain distinct driver mutations with very similar fitness effects independently, the competition between them will be neutral so that none of them can be fixed and they will keep competing. This situation is similar to the phenomenon called "clonal interference" in an asexual population (*Gerrish & Lenski, 1998*).

In actual tumors, driver mutations can not only increase the growth rate but also decrease the death rate. To test the effect of driver mutations decreasing the death rate,

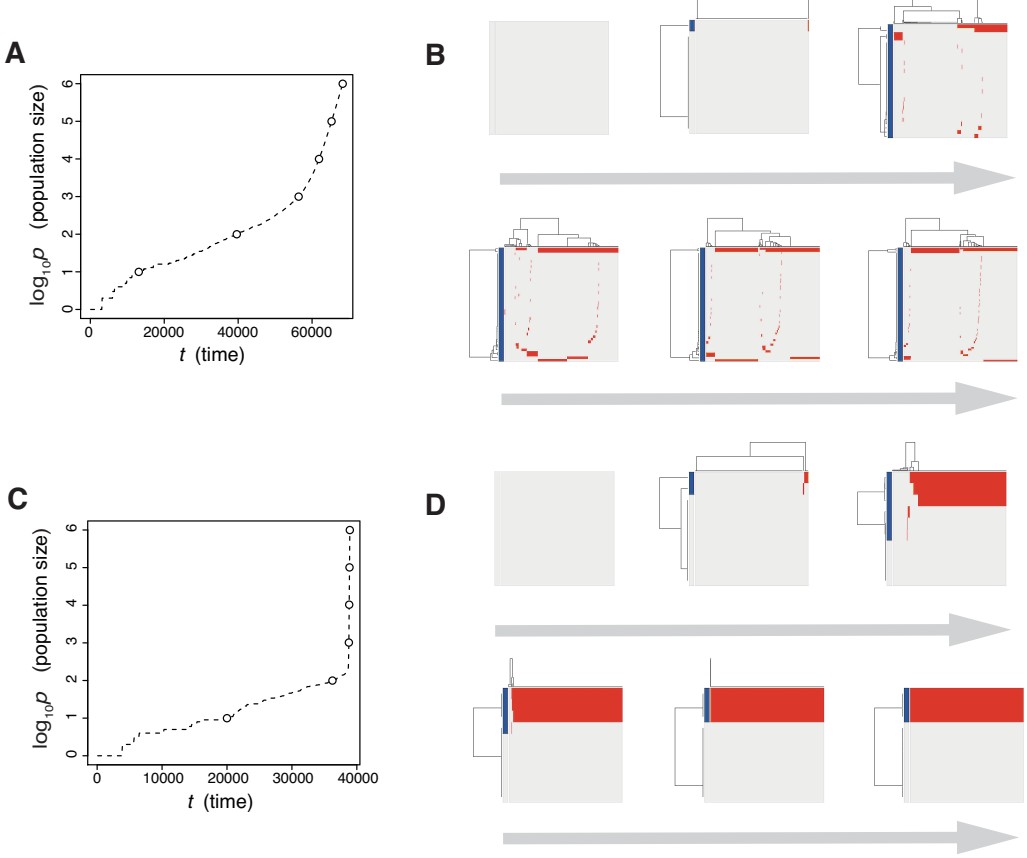

**Figure 3  Time-course snapshots of simulations based on the driver model.** Growth curve (A) and time-course snapshots of mutation profiles (B) simulated from the driver model with $N_d = 3$, $P = 10^6$, $f = 10^{0.3}$, and $m_d = 10^{-1.5}$ (corresponding to parameter setting D in Figs. 2A–2C). Growth curve (C) and time-course snapshots of mutation profiles (D) simulated from the driver model with $N_d = 3$, $P = 10^6$, $f = 10^{0.9}$, and $m_d = 10^{-1.5}$ (corresponding to parameter setting E in Figs. 2A–2C). The time points when snapshots were obtained are indicated by empty circles on the growth curves.

we also created a modified version of the driver model, the "driver-d" model. In the driver-d model, each cell divides with a constant probability $g_0$ and dies with a probability $d$. Driver mutation decreases the cell death probability by $f$ fold: $d = d_0 e^{-n_d}$, where $d_0$ is the base death probability. Moreover, we assumed that cell death occurs only in the case of $p > 1$, to prevent the simulation from halting before clonal expansion. We applied the MASSIVE analysis to the driver-d model to find that, if a high mutation rate is assumed (i.e., $m_d = 10^{-2}$), the driver-branching process is pervasive, irrespective of the strength of the driver mutations (Fig. S2; https://www.hgc.jp/~niiyan/canevosim/driver_d). This observation is presumably ascribed to the fact that a driver mutation that decreases the death rate cannot provide a cell with the strong growth advantage necessary for the linear-replacing process. Even if the mutation rate is low (i.e., $m_d = 10^{-4}$), multiple clones appear after the simulation proceeds to reach a sufficient population size. We also examined

the evolutionary dynamics of the driver-d models with different mutation rates by taking time-course snapshots of the simulations (Fig. S3).

In both the driver and driver-d models, we do not consider spatial information. However, it should be noted that, by simulating tumor growth on a one-dimensional lattice, we demonstrated that the spatial bias of a resource necessary for cell divisions could prompt the driver-branching process (*Niida, Hasegawa & Miyano, 2019*).

## Neutral model

Next, we examined the neutral-branching process by analyzing the "neutral" model, where we considered only neutral mutations that do not affect cell division and death. In a unit time, a cell divides into two daughter cells with a constant probability $g_0$ without dying. Similarly to driver mutations in the driver model, in each cell division, each of the two daughter cells acquires $k_n \sim \text{Pois}(m_n/2)$ neutral mutations. We assumed that neutral mutations acquired by different division events occur at different genomic positions and each cell can accumulate $N_n$ mutations at maximum. In this study, we set $N_n = 1000$, which is sufficiently large that no cell reaches the upper limit, except in a few exceptional cases. The simulation started from one cell without mutations and ended when population size $p$ reached $P$ or time $t$ reached $T$.

The MASSIVE analysis of the neutral model demonstrated that, as expected, the mutation rate is the most important factor for the neutral-branching process (Fig. 4; https://www.hgc.jp/~niiyan/canevosim/neutral_s; note that the neutral model is included by the neutral-s model, which is described below). When the mean number of mutations generated by per cell division, $m_n$, was less than 1, the neutral model just generated sparse mutation profiles with relatively small values of the ITH score, Shannon index 0.05. In contrast, when $m_n$ exceeded 1, the mutation profiles presented extensive ITH, which are characterized by a fractal-like pattern and large values of the ITH score (hereinafter, this type of ITH is referred to as "neutral ITH"). According to these results, it is intuitively supposed that neutral ITH is shaped by neutral mutations that trace the cell lineages in the simulated tumors. Note that the mutation profiles were visualized after filtering out low-frequency mutations. Under the assumption of a high mutation rate, more numerous subclones having different mutations should be observed if we count the mutations existing with lower frequencies.

To verify this speculation, we counted the number of subclones generated from a simulated tumor, while varying the frequency cutoffs for filtering out mutations. Figure S4 shows the plot of the relationship between the number of subclones and the frequency cutoffs. As expected, the results indicate that the simulated tumor presents an increasing number of subclones as the frequency cutoff is lowered. The linearity of the log–log plot demonstrates that the power law is hidden in the mutation profile, consistently with its fractal-like pattern (*Brown et al., 2002*). Note that, although the ITH score does not depend on population size $P$ and the fractal-like pattern shaped in the earliest stage appears to be subsequently unchanged in the time-course snapshots (Fig. 5), these are also because low-frequency mutations were filtered out before visualization; the simulated tumor in fact expands neutral ITH by accumulating numerous low-frequency mutations as it grows.

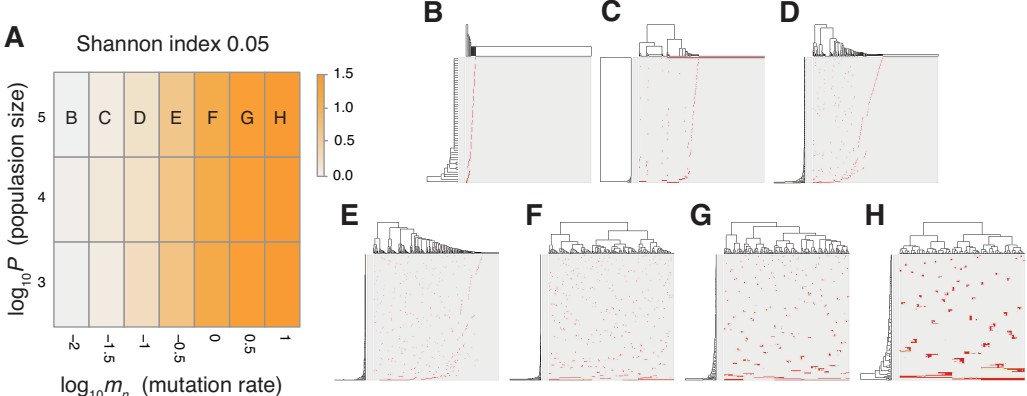

**Figure 4** **Sensitivity analysis of the neutral model.** (A) Heap map obtained by calculating Shannon index 0.05 while changing the neutral mutation rate $m_n$ and the maximum population size $P$. (B–H) Single-cell mutations profiles obtained for seven parameter settings, which are indicated on the heat map in A.

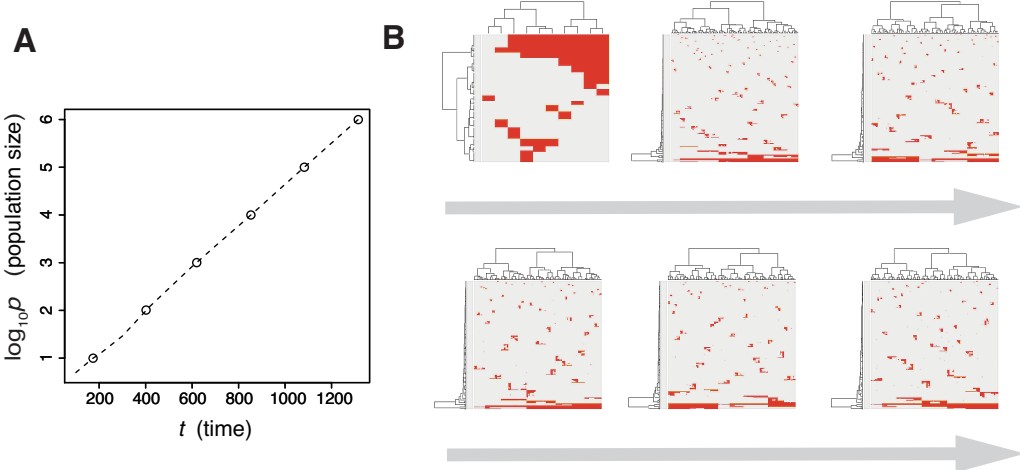

**Figure 5** **Time-course snapshots of simulations based on the neutral model.** Growth curve (A) and time-course snapshots of mutation profiles (B) simulated from the driver model with $P = 10^6$ and $m_n = 10$ (corresponding to parameter setting H in Fig. 4A). The time points when snapshots were obtained are indicated by empty circles on the growth curves.

Thus far, several theoretical and computational studies have shown that a stem cell hierarchy can boost the neutral-branching process (*Sottoriva et al., 2010*; *Solé et al., 2008*), which prompted us to extend the neutral model to the "neutral-s" model such that it contains a stem cell hierarchy (Fig. S5). The neutral-s model assumes that two types of cells exist: stem and differentiated. Stem cells divide with a probability $g_0$ without dying. For each cell division of stem cells, a symmetrical division generating two stem cells occurs with a probability $s$, while an asymmetrical division generating one stem cell and one differentiated cell occurs with a probability $1 - s$. A differentiated cell symmetrically divides

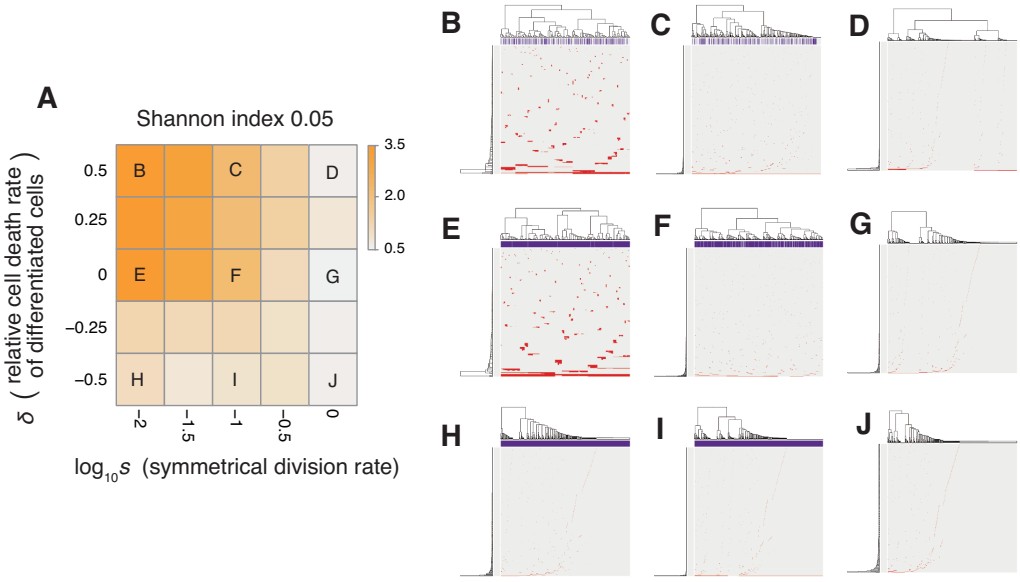

**Figure 6 Sensitivity analysis of the neutral-s model.** (A) Heat map obtained by calculating Shannon index 0.05 while changing the relative death rate of differentiated cells $\delta = \log_{10}(d_0^d/g_0)$ and the symmetrical division rate $s$. The neutral mutation rate $m_n$ and the maximum population size $P$ set to $10^{-1}$ and $10^5$, respectively. (B–J) Single-cell mutation profiles obtained for nine parameter settings, indicated on the heat map presented in A.

to generate two differentiated cells with a probability $g_0$ but dies with a probability $d_0^d$. The means of accumulating neutral mutations in the two types of cell is the same as that in the original neutral model, which means that the neutral-s model is equal to the original neutral model when $s = 0$ or $d_0^d = 0$. For convenience, we define $\delta = \log_{10}(d_0^d/g_0)$ and hereinafter use $\delta$ instead of $d_0^d$.

The MASSIVE analysis of the neutral-s model confirmed that the incorporation of the stem cell hierarchy boosts the neutral-branching process (https://www.hgc.jp/~niiyan/canevosim/neutral_s). To obtain the heat map in Fig. 6A, the ITH score was measured while $d_0^d$ and $\delta$ were changed, but $m_n = 0.1$ and $P = 1,000$ were constantly set. In the heat map, a decrease of $s$ leads to an increase in the ITH score when $\delta \geq 0$ (i.e., $d_0^d \geq g_0$). A smaller value of $s$ means that more differentiated cells are generated per stem cell division, and $\delta \geq 0$ means that the population of the differentiated cells cannot grow in total, which is a valid assumption for typical stem cell hierarchy models. That is, this observation indicates that the stem cell hierarchy can induce neutral ITH even with a relatively low mutation rate setting (i.e., $m_n = 0.1$), with which the original neutral model cannot generate neutral ITH.

The underlying mechanism boosting the neutral-branching process can be explained as follows. We here consider only stem cells for an approximation, because differentiated cells do not contribute to tumor growth with $\delta \geq 0$. While one cell grows to a population of $P$ cells, let cell divisions synchronously occur across $x$ generations during the clonal expansion. Then, $(1 + s)^x = P$ holds, because the mean number of stem cells generated per

cell division is estimated as $1+s$. Solving the equation for $x$ gives $x = \log P / \log(1+s)$; that is, it can be estimated that, during the clonal expansion, each of the $P$ cells experiences $\log P / \log(1+s)$ cell divisions and accumulates $m_n \log P / 2 \log(1+s)$ mutations on average. We confirmed that the expected mutation count based on this formula is well fit with the values observed in our simulation, except in the exceptional cases where the mutation counts reached the upper limit, $N_n = 1,000$ (Fig. S6). These arguments mean that a tumor with a stem cell hierarchy accumulates more mutations until reaching a fixed population size than does a tumor without a stem cell hierarchy. That is, a stem cell hierarchy increases the apparent mutation rate by $\log 2 / \log(1+s)$ folds, which induces the neutral-branching process even with relatively low mutation rate settings.

Similarly, we can also show that the introduction of cell death to the neutral model boosts the neutral-branching process. In the neutral model having a non-zero death rate $d_0$, we estimate that the mean number of cells generated per cell division is $2 - d_0/g_0$. Through arguments similar to the one above, we can also show that the apparent mutation rate is increased by $\log 2 / \log(2 - d_0/g_0)$. Collectively, although the mutation rate is the most important determinant for generating neutral ITH, the introduction of cell death as well as stem cell hierarchy also contribute to the neutral-branching process by increasing the apparent mutation rate.

## Combining the driver and neutral model

We now present the "composite" model that was constructed by combining the driver and neutral model, aiming to reproduce ITH more similar to those in real tumors. In a unit time, a cell divides into two daughter cells with a constant probability $g$ without dying. In each cell division, each of the two daughter cells acquires $k_d \sim \text{Pois}(m_d/2)$ driver mutations and $k_n \sim \text{Pois}(m_n/2)$ neutral mutations. For each type of mutation, $N_d$ and $N_n$ mutations can be accumulated at maximum. For a cell that has $n_d \ (= \sum k_d)$ mutations, cell division probability $g$ is defined as $g = g_0 f^{n_d}$, where $g_0$ is a base division probability. The simulation started from one cell without mutations and ended when the population size $p$ reached $P$ or time $t$ reached $T$. As expected from the MASSIVE analyses of the driver and neutral model that were performed separately, our MASSIVE analysis of the composite model confirmed that, depending on the parameter setting, behaviors of the composite model and the resultant mutation profiles are roughly categorized into the following six classes (Fig. 7; https://www.hgc.jp/~niiyan/canevosim/composite):

- With small $m_d$ and small $m_n$, i.e., with low driver and neutral mutation rates, no evolutionary process involving driver and neutral mutations occurs.
- With large $m_d$, small $m_n$, and small $f$ (i.e., with high driver and low neutral mutation rates, and weak driver mutations), the driver-branching occurs while the neutral-branching process does not occur.
- With large $m_d$, small $m_n$, and large $f$ (i.e., with high driver and low neutral mutation rates, and strong driver mutations), the linear-replacing process occurs while the neutral-branching process does not occur..

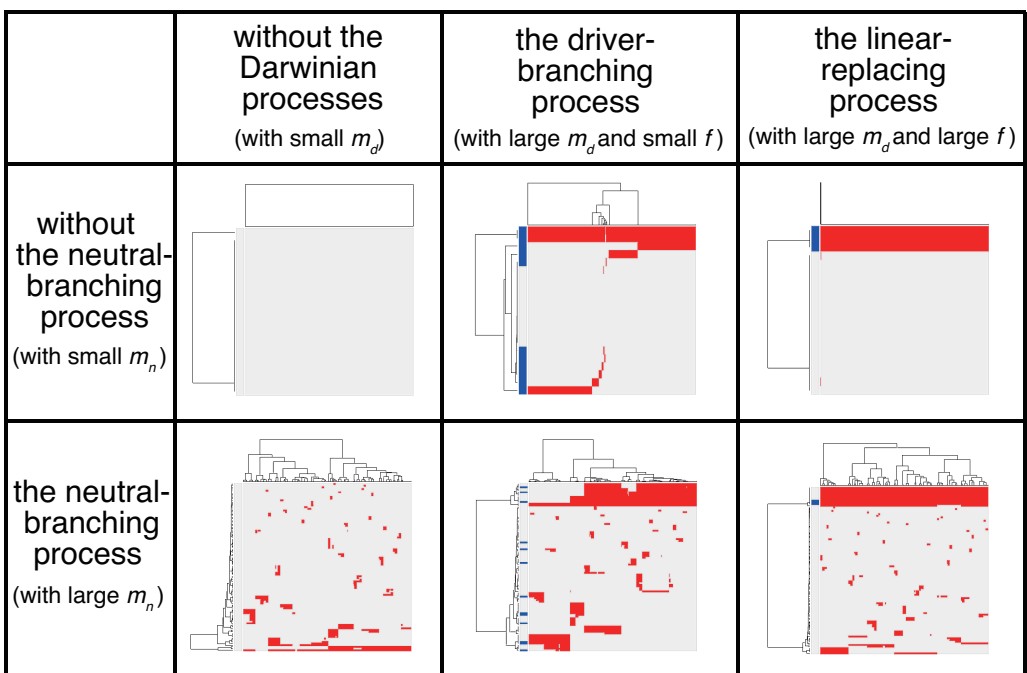

**Figure 7  Six classes of mutation profiles simulated by the composite model.** Our sensitivity analysis demonstrated that, depending on the parameter setting, behaviors of the composite model are roughly categorized into the six classes. Representative mutation profiles of the six classes are presented.

- With small $m_d$ and large $m_n$ (i.e., with low driver and high neutral mutation rates), the neutral-branching process occurs while no evolutionary process involving driver mutations occurs.
- With large $m_d$, large $m_n$, and small $f$ (i.e., with low driver and high neutral mutation rates, and weak driver mutations), the driver-branching and neutral-branching processes occur simultaneously.
- With large $m_d$, large $m_n$, and large $f$ (i.e., with high driver and high neutral mutation rates, and strong driver mutations), the linear-replacing and neutral-branching processes occur simultaneously.

Note that, because tumors having high driver mutation rates must have high neutral mutation rates also, the linear-replacing and driver-branching processes must in general be accompanied by the neutral-branching process. Therefore, the last three behaviors are supposed to constitute the process that can actually occur in real tumors (note that, since different processes work simultaneously and continuously as a series of phases of cancer evolution in real tumors as described below, the situation is not so simple).

## Adding the punctuated-replacing process

Previously, we analyzed multiregion sequencing data of advanced colorectal cancer and precancerous lesions jointly to demonstrated that the evolutionary principle generating ITH shifts from the driver- to neutral-branching process during colorectal tumorigenesis

(*Saito et al., 2018*). We also demonstrated that the number of copy number alterations drastically increases during the progression from colorectal precancerous lesions to advanced colorectal cancer, which prompted us to suspect that the punctuated-replacing process underlies the evolutionary shift from branching to the neutral-branching process (Fig. 1E). To examine this possibility, we additionally incorporated the punctuated-replacing process into the composite model to build the "punctuated" model.

For the models considered thus far, we assumed that a cell can infinitely grow without a decrease in their growth speed. However, it is more natural to assume that there exists a limit of population size because of the resource limitation and that the growth speed gradually slows down as the population size approaches the limit. The limit of population sizes is called the carrying capacity and employed in the well-known logistic equation (*Verhulst, 1838*). By mimicking the logistic equation, we modified the division probability as $g = g_0 f^{n_d}(1 - p/p_c)$, where $p_c$ is the carrying capacity. To reproduce the punctuated-replacing process, we additionally employ an "explosive" driver mutation, which negates the effect of the carrying capacity. After a cell accumulates driver mutations up to the maximum $N_d$, the explosive driver mutation is introduced at a probability $m_e$ after cell division. For a cell that has the explosive driver mutation, the carrying capacity $p_c$ is set to infinite; That is, it is assumed that the explosive driver mutation rapidly evolves the cell so that it can conquer the growth limit and attain infinite proliferation ability.

Next, we searched for parameter settings that lead the punctuated model to reproduce the punctuated-replacing process. The MASSIVE analysis confirmed that, with sufficiently large $m_e$ (i.e., $m_e > 10^{-4}$), the punctuated-replacing process is reproducible in the punctuated model (https://www.hgc.jp/~niiyan/canevosim/punctuated; note that, for simplicity, we omitted neutral mutations by setting $m_n = 0$ in the MASSIVE analysis). We also examined time-course snapshots of simulations conducted with these parameter settings. In the example shown in Figs. 8A and 8B, we observed that multiple subclones having different driver genes coexist; that is, the driver-branching process, with which the neutral-branching process occurs simultaneously, is prominent during the early phase of the simulation. Note that a growth curve plot indicates that, as the population size approaches the carrying capacity, the growth speed slows down; however, the tumor regrows after the appearance of a clone that has acquired an explosive driver mutation. The clone with the explosive driver mutation is then subjected to a selective sweep, which causes subclonal driver mutations in the clone to shift to clonal mutations. Then, only neutral mutations are accumulated as subclonal mutations; That is, ITH is finally generated by the neutral-branching process.

We also found that two subclones having different subclonal driver mutations sometimes appear by obtaining two independent explosive driver mutations (Figs. 8C and 8D). This observation recalls to mind the multiverse model, which was proposed for glioblastoma evolution (*Lee et al., 2017*). The multiverse model is derived from the Big-Bang model, a model for jointly describing punctuated and the neutral-branching process during colorectal tumorigenesis (*Sottoriva et al., 2015*). The Big-Bang model assumes that a single clone explosively expands from a precancerous lesion while generating neutral ITH, consistently with our evolutionary shift model. However, in the multiverse model, it is assumed that multiple subclones are subject to explosive expansion. Collectively,

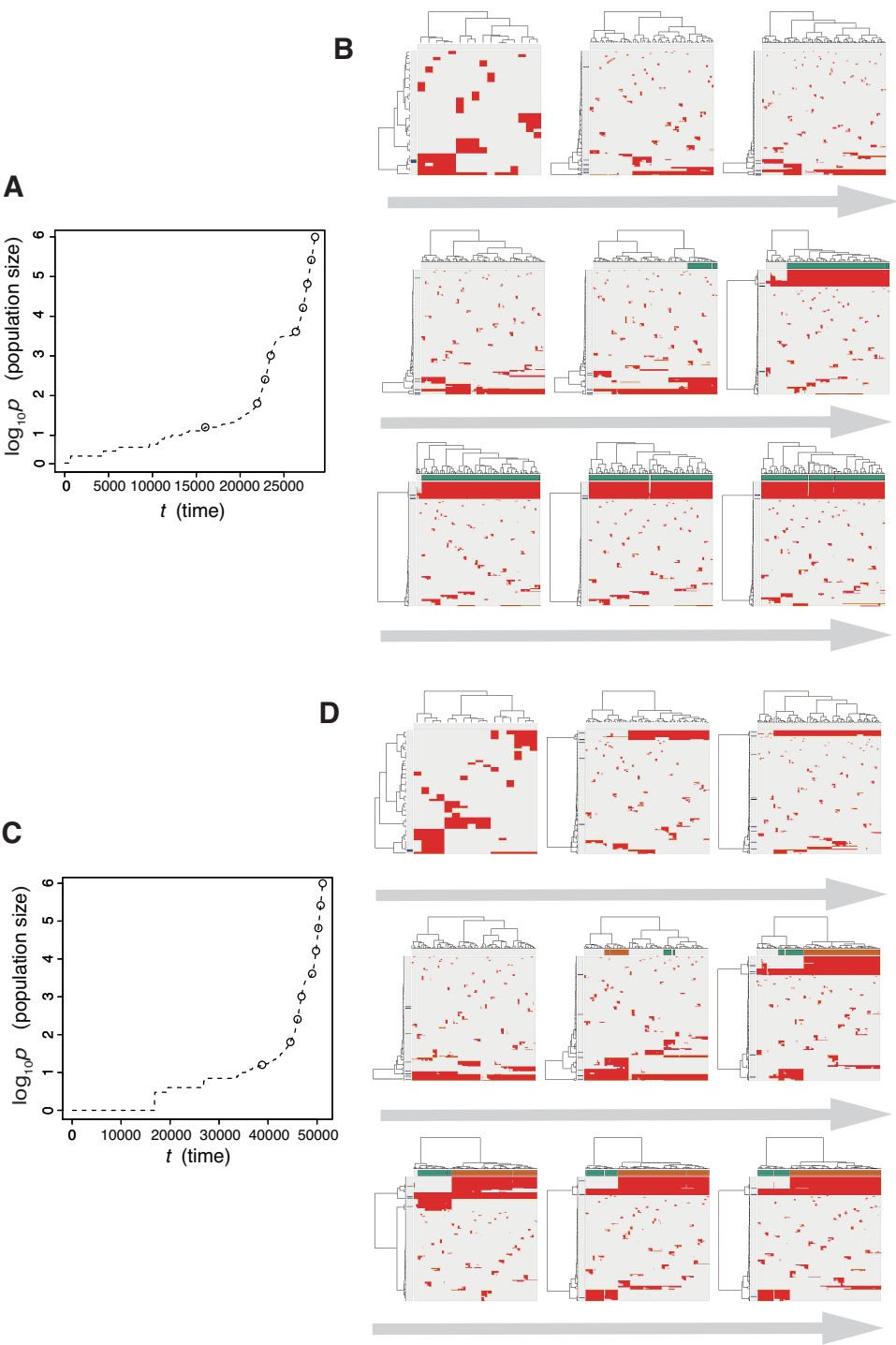

**Figure 8** **Time-course snapshots of simulations based on the punctuated model.** Growth curve (A) and time-course snapshots of mutation profiles (B) simulated from the punctuated model with $P = 10^6$, $p_c = 10^{3.5}$, $m_d = 10^{-1}$, $m_p = 10^{0.5}$, and $m_e = 10^{-4}$. Growth curve (C) and time-course snapshots of mutation profiles (D) simulated from the punctuated model with $P = 10^6$, $p_c = 10^{3.5}$, $m_d = 10^{-1}$, $m_p = 10^{0.5}$, and $m_e = 10^{-3}$. The time points when snapshots were obtained are indicated by empty circles on the growth curves.

our simulation based on the punctuated model not only supports our hypothesis that the punctuated-replacing process underlies the evolutionary shift during colorectal tumorigenesis, but also can reproduce multiple types of punctuated models proposed thus far.

Our simulation based on the punctuated model also demonstrated a dramatic evolution of cancer, during which multiple processes could go on simultaneously and continuously, and we observed different phases along the evolution. Consequently, the mutation profile records the history of the processes such that a series of multiple phases arise with different patterns of mutation profiles. It is possible that we infer the history from the mutation profile at the end point to some degree; for example, the accumulation of clonal driver mutations suggests that the tumor has been subjected to the linear- or punctuated-replacing process. However, our result emphasizes the importance of having a time-series data to fully understand the detailed process behind cancer evolution (*Sato et al., 2019*).

## DISCUSSION

In the 'Results' section, we introduced a family of simulation models that reproduce the four types of cancer evolutionary processes: linear-replacing, driver-branching, neutral-branching, and punctuated-replacing. Our sensitivity analysis of these models successfully identified the conditions leading to each of the evolutionary processes. For example, under the assumption of a sufficiently high mutation rate, the driver-branching process occurs with strong driver mutations, whereas linear evolution occurs with weak driver mutations. However, a major concern about our sensitivity analysis is whether the ranges of parameter values examined is realistic. Although dependent on tumor types, the number of driver mutations were previously estimated as in the low single digits for most tumor types, consistently with our settings for $d$. As the increase in the cell division probability per driver mutation $f$, which is interpreted as the strength of driver mutations, we examined values ranging from $10^{0.1}$ to $10^{1.0}$. Although the value of $f$ has not been the subject to extensive experimental determination, it has been reported that the induction of K-ras$^{G12D}$ in murine small intestine increases growth rate from one cell cycle per 24 hr to one cell cycle per 15 hr, from which $f$ is estimated as $10^{0.204}$ (*Snippert et al., 2014*).

The driver mutation rate $m_d$ and population size $P$ appear to be problematic. Although the driver mutation rate was previously estimated as $\sim 3.4 \times 10^{-5}$ per cell division (*Bozic et al., 2010*), our sensitivity analysis examined values from $10^{-4}$ to $10^{-1}$, which are above the estimated value by orders of magnitude. It should also be noted that, in our simulation, it was assumed that a tumor contains $10^6$ cells at maximum, whereas the number of cancer cells in one gram of tumor tissue is reportedly $10^9$ or one order less (*Del Monte, 2009*). Clearly, for $m_d$ and $P$, the parameter space we examined does not cover those for a real tumor. However, the results of the MASSIVE analysis allow the behaviors of the driver model to be envisioned in a realistic parameter space. When $P$ is small, neither the linear-replacing process nor the driver-branching process occurs. As $P$ increases, we observe the linear-replacing or driver-branching process with smaller $m_d$, although the range of $f$ that leads to the driver-branching process shifts to larger values. Moreover, as
shown by the sensitivity analysis of the neutral-s model, the presence of a stem cell hierarchy increases the apparent mutation rate. Therefore, a real tumor having a stem cell hierarchy apparently should have a higher $m_d$ value. Collectively, it is natural to assume that a real tumor having large $P$ and small $m_d$ can be similarly generated by the linear-replacing or driver-branching process, although, in such cases, the actual value of $f$ might be larger than those that we examined.

The sensitivity analysis of the neutral model showed that neutral ITH is generated if the expected number of neutral mutations generated per cell division, $m_n$, exceeds 1. In a recent report, the estimated somatic mutation rate was given as $2.66 \times 10^{-9}$ mutations per base pair per mitosis. Given that most mutations are neutral on the human genome comprised of $3 \times 10^9$ bases, even a cell division of normal cells generates more than 1 neutral mutation. As cancer cells should have higher mutation rates, which can be further accelerated by stem cell hierarchies, it is reasonable to assume that a tumor in general satisfies the conditions to generate neutral ITH. However, not every tumor necessarily has neutral ITH; neutral ITH is distorted by natural selection if the tumor additionally satisfies the conditions for the driver-branching process, as shown by the analysis employing the composite model.

A highlight of this work is that the punctuated model demonstrated that the punctuated-replacing process triggers the evolutionary shift from branching to the neutral-branching process. For carrying capacity $p_c$ and the probability of acquiring an explosive mutation $m_e$ in the punctuated model, the parameter values that we examined are clearly outside realistic ranges. Similarly to $P$, $p_c$ should take a larger value. Although it cannot easily to be experimentally determined, $m_e$ also appears to be overestimated; although the human body in fact potentially harbors numerous precancerous lesions (*Brunner et al., 2019*; *Yokoyama et al., 2019*), which are assumed not to have acquired explosive driver mutations yet, only a tiny fraction of cases progress to advanced stages by acquiring explosive driver mutations. However, it is intuitively understandable that the behaviors of the punctuated model, as well as of the driver model, are not dependent on precise values of these parameters, and in our opinion, our analysis is sufficient to provide a semi-quantitative understanding of cancer evolution.

The models we introduced in the Results section can be described collectively as the unified model, a formal description of which is provided in the 'Materials & Methods'. The unified model is very simple but sufficient to reproduce the linear-replacing, driver-branching, neutral-branching, and punctuated-replacing processes. Of course, the unified model harbors many limitations, which should be addressed in future studies. Our current version of the model completely ignores spatial information, which potentially influences evolutionary dynamics. Recently reported studies have shown that spatial structures regulate evolutionary dynamics in tumors (*Noble et al., 2019*; *West et al., 2019*). We also determined that resource bias prompts the driver-branching process, by simulating tumor growth on a one-dimensional lattice (*Niida, Hasegawa & Miyano, 2019*). Moreover, *Iwasaki & Innan (2017)* recently developed a realistic simulator called tumopp to show that the three-dimensional pattern of ITH is affected by the local cell competition and asymmetric stem cell division. Although our model assumed that driver mutations independently have

effects of equal strength, different driver mutations should have different strengths and might work synergistically (*Castro-Giner, Ratcliffe & Tomlinson, 2015*). Similarly, although we assumed that the punctuated-replacing process occurs only once in the course of cancer evolution, it is possible that a tumor is confronted with different types of resource limitations during the tumor progression and undergoes the punctuated-replacing process multiple times to conquer them (*Aktipis et al., 2013*).

## CONCLUSION

Although the unified model harbors the above-described limitations, the application of sensitivity analysis to the model has successfully provided a number of insights into cancer evolutionary dynamics. In our opinion, the unified model serves as a starting point for constructing more realistic simulation models to understand in greater depth the diversity of cancer evolution, which is being unveiled by the ever-growing amount of cancer genomics data.

## ACKNOWLEDGEMENTS

We thank Hiroshi Haeno and Watal M. Iwasaki for helpful discussions.

### Funding

This research was supported by AMED under Grant Number JP17cm0106504h0002; MEXT as Priority Issue on Post-K computer (Supercomputer Fugaku) (Project ID: hp170227); and JSPS as Grant-in-Aid for Scientific Research on Innovative Areas (15H0912) and Grant-in-Aid for Scientific Research (JP15H05707 and JP19K08123). The funders had no role in study design, data collection and analysis, decision to publish, or preparation of the manuscript.

### Grant Disclosures

The following grant information was disclosed by the authors:
AMED: JP17cm0106504h0002.
MEXT as Priority Issue on Post-K computer (Supercomputer Fugaku): hp170227.
JSPS as Grant-in-Aid for Scientific Research on Innovative Areas: 15H0912.
Grant-in-Aid for Scientific Research: JP15H05707, JP19K08123.

### Competing Interests

The authors declare there are no competing interests.

### Author Contributions

- Atsushi Niida conceived and designed the experiments, performed the experiments, analyzed the data, prepared figures and/or tables, authored or reviewed drafts of the paper, and approved the final draft.

- Takanori Hasegawa analyzed the data, prepared figures and/or tables, and approved the final draft.
- Hideki Innan performed the experiments, authored or reviewed drafts of the paper, and approved the final draft.
- Tatsuhiro Shibata, Koshi Mimori and Satoru Miyano conceived and designed the experiments, authored or reviewed drafts of the paper, and approved the final draft.

## Data Availability

All the results in this study can be interactively explored in the MASSIVE viewer on our website: https://www.hgc.jp/~niiyan/canevosim.

The simulation code is available at: https://github.com/atusiniida/canevosim.

## Supplemental Information

Supplemental information for this article can be found online at http://dx.doi.org/10.7717/peerj.8842#supplemental-information.

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
