# Peer review of "A unified simulation model for understanding the diversity of cancer evolution"

_PeerJ, doi:10.7717/peerj.8842_

## Round 0.1 · original submission · Major Revisions

Your manuscript has now been seen by two reviewers. You will see from their comments below that while they find your work of interest, some major points are raised. We are interested in the possibility of publishing your study, but would like to consider your response to these concerns in the form of a revised manuscript before we make a final decision on publication. We therefore invite you to revise and resubmit your manuscript, taking into account the points raised. Please highlight all changes in the manuscript text file.

Reviewer 1 ·

Basic reporting

The article describes a simulation framework and software to model cancer evolution under different modes.

There are places where the English could be improved. More specifically, the introduction is fine, but in the Results section the tenses are inconsistent (e.g., switching from past to present in l. 83 and 84) and some sentences are misleading (e.g., l. 83 “each agent corresponds to each cell in a tumor”, it sounds like an agent represents several cells). There are also some formatting issues that need to be corrected. The citations are sometimes not well embedded in the text (e.g., l. 36, the author name should be between parentheses), which makes the reading more difficult. Please correct.

I find the introduction not sufficiently clear, lacking definitions of the basic phenomena described in the paper. In particular, the branching evolution model is never really defined. Also, the sentence l. 42 is misleading; “[...] intratumor heterogeneity (ITH) [...] results from branching evolution” makes it sound like ITH always results from branching evolution, which is in contradiction with the rest of the introduction, and the evolutionary literature.

The citations are insufficient (less than 30 citations is quite small for a methodological paper grounded on a rich literature from evolutionary biology and cancer). Firstly, I think citations are lacking to backup some statements. l. 37, the “linear evolution” hypothesis is stated without reference; I can suggest citing Nowell, P. C. The clonal evolution of tumor cell populations. Science 194, 23–28 (1976), but the authors might have another one in mind. Secondly, and more importantly, in general I think more credit must be given to early work or at the very least, reviews of such work. For example, regarding the usefulness of next generation sequencing l.40 to study cancer evolution, I think credit should be given to earlier studies or at least earlier reviews such as Yates, L. R., & Campbell, P. J. (2012). Evolution of the cancer genome. Nature Reviews Genetics, 13(11), 795. Also, because the punctuated equilibrium hypothesis dates from the 1970s, it sounds weird to only cite very recent work. Finally, I think more studies of mathematical and computational models of cancer evolution should be described, and the previous results should be discussed (e.g., what of Beerenwinkel, N., Schwarz, R. F., Gerstung, M., & Markowetz, F. (2014). Cancer evolution: mathematical models and computational inference. Systematic biology, 64(1), e1-e25.)

The figures are of good quality. I think the axes labels on Fig. 2 are currently barely readable and should be increased though. The sharing of all the code on github is commendable. I believe the papers follows very good standards in terms of reproducibility.

The article is self-contained.

Experimental design

In my opinion, the research topic is quite interesting and in the scope of the journal.
The research question is well defined. A gap in the current knowledge is clearly stated, but although I agree that there is indeed a need for a simulation software to tackle the issues mentioned in the introduction, I think that there is not enough references to the literature on mathematical modelling of cancer evolution to justify why we also need an additional mathematical model in addition to the ones already developed.

I have concerns regarding the methods. I do not agree with the fundamental assumptions of the model. In the driver model, each mutation increases the cell division rate. Although such mutations accelerating the rate of cell division probably exist, this is not what is commonly referred to as Darwinian selection in the evolutionary literature, where a positively selected mutation leads to more “offsprings” due to several phenomena, among which the most common is increased survival. An increased cell division rate indeed would lead to more offsprings per unit of time, and thus can be considered as a fitness increase and one possible type of drivers, but reducing all drivers to such a model is a very strong assumption that would need to be backed up either by very strong empirical data in the cancer literature. For my part, I believe that this assumption fatally reduces the scope of the model; for example, it excludes resistance mechanisms against important therapeutic strategies such as immune checkpoint blockade, where driver events leading to the evolution of an immune escape phenotype actually increased survival of cells by reducing predation from the immune system, without ever acting on the cell division rate. Also, I think the assumption that most tumors undergo numerous events increasing the cell division rate needs to be justified.

I find that the results of the driver and neutral models do not add much to the literature, so emphasis on them should be reduced and proper literature should be cited (this basic effect of mutation and selection has been long described in population genetics, and can already be simulated forward in time for example using python package Simupop). The model with stem cell hierarchy is the true novelty of the model.

The punctuated evolution model is also problematic, as what it actually models (sudden changes in carrying capacity) do not correspond to what is classically understood as punctuated evolution.

I find that the sensitivity analysis is good but too qualitative. For example, l. 209-220, 6 qualitative behaviors are stated but without any comment on the range of parameters that lead to these behaviors (what is a small md, exactly?). Also, it ignores the issue of the transitions between these behaviors: what happens when f is intermediate? how sharp is the transition between the small and large f behaviors? I think this is especially important to describe because most of the behaviors described are expected and thus the input of the simulations proposed is to get actual quantitative measures.

Validity of the findings

Because of the concerns raised raised above, I do not think that the results are sufficiently sound. Indeed, statements like "under the assumption of sufficiently
high mutation rates, branching evolution occurs with strong driver mutations" are misleading, because "strong driver mutations" refer to the very specific model presented here, and thus is an extrapolation of the results. Also, apart from the results from the neutral-s model and the punctuated model, the results constitute a repetition of textbook population genetic results (e.g., the joint effect of mutation and selection are well-known), so I suggest again reducing the emphasis on them.

Additional comments

I think that the authors should revise the fundamental assumptions of their model to make it more general; as it stands, I think its description is misleading as it sounds very general while actually describing a very specific scenario, and probably not the most common. Also, although novelty is not a publication criterion here, more citations are needed to give credit to the abundant evolutionary literature and other attempts at modelling cancer evolution, and to justify the assumptions of the model.

·

Basic reporting

Manuscript presents a driver model to study cancer evolution, mainly analysing clonality and branching structures of oncogenesis. The idea is well defined and executed. However, this strategy is not novel (as stated in the abstract), as models with the same basic rationale were already published (MInussi, Henz and Oliveira, Cancer Research, 2019). However, the technicalities of this model and the model presented in the present manuscript are different, which makes the present model worth publishing. It would be interesting if the key findings of this work could be confirmed by the previously published model.
In general the role of cell death is undermined, as in most tumours, cells proliferation coexists with cell death. In addition, immune surveillance also adds growth pressure, mainly mediated by cell death. Please discuss the role of cell death more clearly in all 4 models of cancer evolution.

The Figure models of evolutionary trees presented throughout the manuscript makes it very difficult to read. Consider other forms of depicting the profiles (Fig 1D), or at least make the lines of the clustering stronger. Additionally, identifying what would be considered a clone in these profiles would improve clarity of the figure.
Heat maps would be easier to visualize if the legends instead of symbols had also a description of their meanings.

Minor: There are some typos, especially in the abstract.

Experimental design

no comment

Validity of the findings

Findings are important, but are limited by the limited data available of cancer evolution. This limitations are, however, clearly stated in the discussion.

Additional comments

no comment

---

## Round 0.2 · accepted · Accept

Thank you for the detailed response letter. We are delighted to accept your manuscript for publication.

Reviewer 1 ·

Basic reporting

I am very pleased with the additional references and improvements in English in this revised version.

Experimental design

I am pleased with the clarifications of the research question and models, and the addition of new models.

Validity of the findings

I find the additional models, and the additional information about hypotheses of the model satisfactory.

Additional comments

I find the manuscript much improved. The efforts of the authors to rethink the manuscript and models are very welcome. The framework developed by the authors could be quite useful to explore evolutionary scenarios.